A robust detect and describe framework for object recognition in early childhood education

Lv Lan
Yao Suhui yaosuhui@zzpec.edu.cn
Preschool Education Institute, Zhengzhou Preschool Education College , Zhengzhou , China
Asif Muhammad
Electronic publication date: 2025 Sep 17
Publication date: 2025
Volume: 11
Electronic Location ID: e3080
Received 2025 Apr 13; Accepted 2025 Jul 4
Copyright: © 2025 Lv and Yao
Copyright year: 2025
Copyright holder: Lv and Yao
License: This is an open access article distributed under the terms of the Creative Commons Attribution License, which permits unrestricted use, distribution, reproduction and adaptation in any medium and for any purpose provided that it is properly attributed. For attribution, the original author(s), title, publication source (PeerJ Computer Science) and either DOI or URL of the article must be cited.
License URL: https://creativecommons.org/licenses/by/4.0/

Keywords: Object detection, Contents recognition, Image to text, Preschool education, Elementary training

Funding: The authors received no funding for this work.

==============================
Preschool education plays a vital role in the harmonious development of an individual. Understanding basic shapes, colors, and letters at an early age lays a strong foundation for academic excellence and emotional growth. At an early childhood stage, the skills of spatial reasoning and problem-solving can be developed by recognizing and comprehending the depicted objects. By exploring deep learning technology, this article presents a cognitive enhancement framework for recognizing nested objects. With cutting-edge models, such as You Only Look Once (YOLOv8) and Visual Geometry Group (VGG16), objects and intra-objects are detected. For semantic description, the neural network model, specifically long short-term memory (LSTM), is exploited, preceded by precise object recognition. The framework is implemented in Google Colab with the prominent packages of Ultralytics, PyTorch, and OpenCV. The models are trained and tested by a custom dataset: PreEduDS. The results of the systematic evaluation suggest that the framework has widespread applicability. A promising accuracy score of 94.4% is obtained for object recognition and 96.5% for predicting precise semantic textual description. The proposed system is well-suited for enhancing preschool education and training based on augmented reality (AR) applications.

Introduction

Nursery education is a crucial stage in a child’s lifelong cognitive development. Elementary education aims to mold the thinking and socio-emotional abilities of the learner through understanding and interaction with objects. Traditionally, teachers relied mostly on static tools, such as flashcards, magazines, and textbooks, which were inadequate for fully engaging learners. The static instructional material used in conventional teaching not only encourages rote learning but also limits creativity (Gyekye-Ampofo, Opoku-Asare & Andoh, 2023). Additionally, the development of holistic cognitive ability was always hindered due to a lack of sensory-rich resources (Laghari, 2024). Computer-based technologies provide a more dynamic and interactive learning experience. Moreover, educational software and multimedia resources are well-suited for the deeper engagement of young learners (Haugland, 2000; Wang et al., 2022), whereas digital storytelling is effective in enhancing cognitive development (Xiong, Liu & Huang, 2022). Moreover, such tools facilitate instructors in offering content tailored to the child’s preference and pace (Plowman & Stephen, 2003).

In all preschool pedagogical software, digital images containing letters, shapes and objects are analyzed. Hence, in the teaching-learning process, the detection, recognition, and textual representation of objects are necessary. Although algorithms have been proposed in the literature for detecting and labeling objects in input images, challenges such as occlusion and spatial hierarchies still need to be overcome. With the emergence of deep learning technologies, multi-class object recognition with promising accuracy has become possible. A deep learning-based image processing system is the ideal solution for robustly recognizing objects and accurately presenting image content, which is the focus of this research. Learning and development in thinking, feelings and social behaviors begin in preschool. In the beginning, kids are laying the groundwork for abilities in speaking, solving problems and understanding space. A great method to support these abilities is to use learning that involves children seeing and interacting with objects, shapes and letters. AI is starting to show encouraging ways it can influence education and transform traditional classrooms as technology advances. In particular, object detection and semantic description systems can help enhance the quality of education for preschool learners by offering engaging, personalized, and interactive ways to learn.

This study presents the design and implementation of a practical framework for accurately detecting objects in input images and precisely presenting their contents in textual form. The system operates in two phases: the objects extraction phase (OEP) and the object description phase (ODP). In the OEP, objects and intra-objects are detected in input images using the You Only Look Once (YOLOv8) model (Chen et al., 2025). Second phase: ODP, object recognition is performed by VGG16 (Simonyan & Zisserman, 2014), whereas semantic description in textual form is achieved by the long short-term memory (LSTM) (Hochreiter & Schmidhuber, 1997). A custom dataset, PreEduDS (Preschool Education Dataset), comprising 730 images, is created and used for training, validation, and testing of the models. Images of the dataset, containing objects and nested objects, are gathered from elementary school books available in both soft and hard formats. Image augmentation is performed to increase the variation of images, followed by proper annotation. The YOLOv8 model in OEP exploits a single convolutional neural network (CNN) to split the image into a grid of cells. The class probability, in addition to the likelihood of a distinct object, is predicted for each cell. As a post-processing step, non-maximum suppression (NMS) is performed to prevent overlapping objects. In ODP, the VGG16 is employed as an encoder to extract features of objects from the pixel values, which are normalized in the range of 0 to 1. The recurrent neural network, LSTM, is used as a decoder to generate descriptive text based on the features of the encoder. For easy understanding of the objects carried in an image, object names are displayed at the detection stage alongside descriptive text for effortless comprehension. The working schematic of the framework is shown in Fig. 1.

Figure 1 Schematic of the working of the D&D framework.

The framework is assessed from various perspectives, such as object detection in OEP and generation of seamless sequences after object recognition in ODP. Almost all objects and nested objects are accurately detected by the YOLOv8 model in the first phase, with an accuracy of 96.4% and an F1-score of 0.95 obtained in this phase. However, a comparatively low accuracy of 0.92% with an F1-score of 0.93 is reported when evaluating the performance of VGG16 in the second phase. As a whole, the framework applies to computer-based training (CBT) in general and to the emerging computer-based early years education (CBEY) in particular.

The remaining article is organized into four sections. “Literature Review” discusses related studies, and “Methodology” presents the methodology. Evaluation and result analysis are presented in “Implementation and Evaluation”, while the conclusion with future direction is covered in “Conclusion and Future Work”.

Literature review

Elementary education is of utmost importance in shaping the cognitive and psychomotor skills of young learners. It is evident from the literature that elementary education enhances an individual’s readiness for formal schooling and sharpens their basic skills in problem-solving and language learning (Liao et al., 2024). During this formative period, emotional and cognitive skills are molded; therefore, systematic teaching is required to enhance the learning outcomes (Shonkoff & Phillips, 2000). With the advancement of computing technology, systems have been devised to improve engagement among children. Interactive games, visuals, and animations are designed to support the harmonious growth of preschool children (Shonkoff & Phillips, 2000). To stimulate curiosity and sharpen critical thinking, virtual reality and augmented reality (AR)-based immersive learning platforms have also been proposed (Yi, Liu & Lan, 2024). Such technologies are beneficial for customized learning, particularly in addressing the issues of pace and style for each learner (Abrar et al., 2019).

The emerging technology of deep learning, particularly text generation, has revolutionized the domain of education (Ye et al., 2024; Liu et al., 2024). To better associate objects and recognize their real-world counterparts, computer vision projects play a crucial role (Simonyan & Zisserman, 2015). To better understand everyday objects, systems for object detection and recognition have been proposed (Redmon et al., 2016). For language development in early years, researchers are suggesting a text-generation system (Ye et al., 2024). To foster cognition, systems for detecting objects and generating descriptive captions are in use (Liu et al., 2016). Stance, an artificial intelligence (AI)-based system (Liu & Brailsford, 2023), interactively engages children with visual tools, whereas Chung et al. (2025) have successfully created a personalized learning environment using AI gaming. However, as stated in Chung et al. (2025), Ghandi, Pourreza & Mahyar (2023), and Lin et al. (2017), several issues arise in the detection and prediction of precise captioning in such a system. For accurate labeling, the pre-trained masked language models are proposed in Hossain et al. (2019), Shi, Dao & Cai (2025), Wang et al. (2025). However, these systems lack task-specific adaptation.

Recent studies indicate that technology is growing in importance for early learning. AR and virtual reality (VR) are being integrated into preschool classrooms to provide students with engaging experiences. For instance, AR is being explored to engage young children in learning about space and problem-solving. At the same time, VR is being utilized to support social and emotional skills in early education. They help create better learning experiences for children by letting them explore educational content interactively.

Additionally, deep learning technologies are proving very helpful for recognizing objects and generating text, helping to advance education at the beginning of a child’s learning journey. New developments in AI for personalized learning in preschool demonstrate that AI systems tailor learning content to meet each child’s individual needs. Because CNNs and recurrent neural networks (RNNs) are designed for image and text processing, they are well-suited to object recognition and generating descriptive texts in learning software (Shi, Hayat & Cai, 2024; Liu et al., 2024; Li et al., 2025). Therefore, there is a pressing need for a robust system that not only effectively detects objects but also presents the contents of images in a simple, textual form. Therefore, this research work is proposed with the intent of having an applicable framework that detects, recognizes, and describes the contents of images.

Methodology

Needless to say, preschool education has a substantial impact on an individual’s cognition development. Learning through visual aids is the widely accepted early education method to enhance children’s observational and problem-solving skills. The detection of nested objects in images is an active area of research. This study aims to design a framework that utilizes cutting-edge deep learning and transfer learning approaches to detect and describe objects in an input image accurately. The system not only helps students learn objects, digits, and alphabets easily but also enables teachers to make pedagogy more engaging and effective. The PreEduDS data was created by collecting 730 images from common online teaching materials and textbooks used for preschool learning. The images are drawings of objects, letters, digits, and shapes that are usually introduced in preschool. The reason for choosing these resources is that learners can explore different topics and fundamental educational themes. Information used in the dataset was obtained from both digital and printed learning materials.

Figure 1 lays out the main stages of the proposed framework we are considering. Before proceeding, the dataset is first resized, normalized, augmented, annotated, and tokenized, ensuring the images are ready for object recognition and text creation. When we resize the pictures, we make them all the same size. Normalization takes all pixel values and changes them to a uniform range. Augmentation adds more variety to the data. Annotation names everything in the images and tokenization changes the annotated text into input tokens for the text generation. During the second step, the OEP, objects within the images are identified, and overlapping detections are removed to prevent repeated findings. Once the objects pass the RS stage, they are sent to the ODP, where the recognized objects are further processed and textual descriptions are developed. At the output stage, you will see the detected objects and their descriptions, also known as tags, such as ‘A robot and apple.’ The sequence in the framework helps turn data into meaningful output with proper recognition and text. Details about the framework are presented in the following subsections.

Preprocessing

The PreEduDS dataset, comprising 730 images, is preprocessed for practical training. The standard Roboflow web app (Liu et al., 2025) is utilized to annotate the photos. The annotation is saved in the YOLOv8 format after normalization into the range of 0 and 1—the classes. A named file is created, containing the names of all the objects, and is made a part of the dataset. The dataset was annotated to include the relevant object classes for detection and recognition. While the dataset is diverse in terms of object types, potential biases may exist due to the sources of the images, as they primarily come from digital and physical resources commonly used in educational settings. These resources may not fully represent all cultural contexts or image variations found in a broader global context. Therefore, the dataset may have a bias toward the specific teaching materials available in certain educational systems. The steps performed for preprocessing are as follows.

Image resizing

To bring all PreEduDS images to a fixed standard size, image resizing is performed while preserving pixel information. A standard scaling factor S is applied to image I of height h and width w to get a resized height h′ and width w′,

(1) S=h/wh′/w′.

Resizing is followed by bicubic interpolation ( Ibicubic) to obtain accurate pixel intensity. Figure 2 shows this resizing process.

(2) I′=Ibicubic(I,h′,w′).

Figure 2 Image resizing followed by bicubic interpolation.

Normalization

To bring the pixel values of images into a standard range of [0,1], the method of z-score normalization is employed. A normalized image IN is obtained from an input image I of ‘c’ number of channels as,

(3) IN=Ic−μσ

where,

(4) μ=1N∑c⁡Ic

(5) σ=1N∑c⁡(Ic−μ)2.

Augmentation

To synthetically expand the variability of images, augmentation is performed over the dataset. The two standard transformations- rotation ( τR) and flipping ( τF) are performed to get the augmented images IAug. The original labels of images are retained during the production of augmented images.

(6) ImgAug=τR(I,θ)

(7) ImgAug=τF(I,dr)

where θ is the rotation angle and dr The direction of flipping (vertical, horizontal). The augmentation of an input image is shown in Fig. 3. The training data was modified using image augmentation to avoid overfitting. We first rotated a portion of the images, typically by ±30 degrees, to enable the model to detect items at various angles. Additionally, we flipped some images to help it cope with flipped versions of objects. The additional information enables the model to comprehend unseen objects and various angles and placements of objects.

Figure 3 Input image (Orig.) with its augmentation versions.

Annotation

Annotation is performed to label objects in the dataset. For n objects of an image I, the object Om is annotation as,

(8) Om={Cl,Bx,By,Bw,Bh}

where Cl represents the class label, Bx, By the coordinates of the bounding box around the object, and Bw, Bh is the width and height of the box. The text file in YOLOv8 format of the image- If containing n objects is presented as,

(9) If=[Cl1Bx1By1:::ClnBxnBynBw1:BwnBh1:Bhn].

The process of annotation is performed for each I∈PreEduDS.Train with the Roboflow App. Once annotation is completed, the labels with caption files are integrated into PreEduDS, as shown in Fig. 4.

Figure 4 The process of annotation and updation of the PreEduDS.

Tokenization

The neural network models encode I∈PreEduDS into a d-dimensional feature vector Fv;

(10) Fv=Encodes(I),Fv∈Rd.

For label L of object O in image I having words- w1 to wn, is given as,

(11) L={w1,w2,….,wn}.

To have the start and end of each label, a starting word STR and trailing word END is added,

(12) L={Str,w1,w2,….,wn,End}.

With a tokenizer- Tr, each word wi is mapped into the token ti, to have the corresponding label in tokens LT within a range of 1 to vocabulary size V,

(13) ti=Tr(wi)ti∈{1,2,….|V|}

(14) LT={t1,t2,….tn}T∈Zn.

With an embedding matrix M, the token ti is embedded into a dense vector

(15) ei=M[ti],ei∈Rk

the label is thus represented as a dense embedded vector of dimension k for onward processing,

(16) ML={e1,e2,…en}ML∈Rk.

Serving as a decoder, the LSTM generates a predicted token ( Pt) for the feature vector at the hidden state- hti and time ti,

(17) Pt=Decoder(ti,Fv,hti).

OEP

In the first phase, objects and intra-objects are detected in the input image by the prominent YOLOv8 neural network. Unlike other CNN models that require multiple passes over an image for object identification, the YOLO model is exploited in the first phase to extract objects efficiently. The model identifies and classifies objects in a single pass by treating the input image as a grid of cells. OEP comes first and feeds the information gathered into the ODP. Detection and location of objects in the image are accomplished with YOLOv8 in OEP. Following detection, the bounding boxes are transmitted to ODP, where VGG16 identifies the objects, and the LSTM model generates write-ups based on the characteristics of the objects detected by VGG16. Because detection leads directly to description, things are not only recognized but also provided with meaningful context-specific text. While examining which part of an object lies in which cell, multiple objects are simultaneously identified. Initially, the feature pyramid network, containing convolutional layers, is established to process the input image at multiple resolutions for detecting various objects. The reason for having a feature pyramid network is that it can handle objects of any size. Using a top-down network with lateral links, it assembles a pyramid of multi-level features, allowing the network to detect objects of any size. For this reason, YOLOv8 demonstrates better results for objects that may appear as small or large in an input image.

The LSTM model describes the features identified by the object recognition process. YOLOv8 first detects objects and VGG16 then finds meaningful features from those objects before the LSTM uses these features as input. After that, the LSTM works as a decoder to turn the features into a sequence of words. The features from VGG16 help form a detailed explanation of the objects found in the image. LSTM processes information by converting the sequence of features from VGG16 into an output sequence (descriptive text). The model generates sentences about objects and outputs them as text descriptions. When object recognition and LSTM are used, the framework can identify and describe objects, providing a comprehensive explanation of what is in the image. In the feature maps (feature pyramid ( ρx)). The fine details for simple and small objects are extracted by ρ1 whereas semantic features by larger receptive fields ( ρ2−ρ5). If Fi and Fo are for input and output features, w the weights of convolutional and b the bias, the process is represented as:

(18) Fo=w∗Fi+b.

By the neck of YOLOv8, feature refinements and upsampling (U) are performed using single (C) and two convolutional layers (C2f), as shown in Fig. 5, demonstrating the working of the YOLOv transformer. The goal is to enhance the spatial resolution of feature maps and to harmonize the features of the deeper and shallower layers. For bilinear up-sampling, the interpolation is performed as:

(19) Fu(x,y)=∑k=01⁡∑l=01⁡IWk,l∗Fi(k′,j′)

where IW is the interpolation weight.

Figure 5 The YOLOv8 architecture where Pn represents the feature pyramid levels, C single convolution layer, C2F two convolution layer with feature fusion, and U represents the un-sampling operation.

The head of YOLO predicts the objects’ class using softmax after tracing its location in the image map using the probability- Ci of the ith class:

(20) Ci=exk∑l=1C⁡exl

where xk and xl represents the logit of the kth and lth classes associated with raw scores.

The binary cross entropy loss is computed by using the entropy loss ( EL) for each single prediction to ensure that a cell of the grid contains an object or not. Where t is the ground truth and p is the predicted probability.

(21) EL=−[y.log⁡(p)+(1−y).log⁡(1−p)].

To avoid the possibility of overlapping and redundant cell detection, non-maximum suppression (NMS) is performed as a post-prediction operation. NMS enhances raw predictions and ensures accurate detection of objects. With NMS, the technique removes extra boxes that appear for the same object by choosing only the one with the top score. Boxes that share large regions are marked for removal, as this leads to more precise object detection. By filtering out false detections and saving only the important ones, NMS leads to a more accurate and no-repeated detection outcome. For the highest scoring cell, for every cell, the intersection over union ( IU) is computed from the area of overlap ( AO) and area of union ( AU); as shown in Fig. 6,

(22) IU=AOAU

If IU>0.5, the cell overlapping cells are deemed redundant and are avoided for onward processing.

Figure 6 Object detection with NMS to avoid false detection and overlapping.

In our framework, the interaction between the YOLOv8, VGG16, and LSTM models occurs in two main phases. In the first phase, YOLOv8 is used for object detection, where it identifies and locates objects and intra-objects in the input image. YOLOv8 works by dividing the image into a grid, and each grid cell predicts class probabilities and bounding box coordinates. Once objects are detected, their bounding boxes are passed on to the second phase. In the second phase, the VGG16 model utilizes the detected objects for feature extraction. It processes the images of detected objects, extracting relevant features from the pixel values. These extracted features are then fed into the LSTM model. The LSTM serves as a decoder, generating a semantic description of the objects in the image. The LSTM model uses the feature representations from VGG16 to predict a sequence of words, effectively generating a textual description of the objects present. This interaction between detection, feature extraction, and text generation enables our framework to not only detect objects but also describe them in a meaningful way.

ODP

In the second phase, a meaningful descriptive text about an input image is generated after recognizing objects within the image. The CNN-based deep learning model, VGG16, is utilized for object recognition, and the LSTM is employed for text generation. The classes of objects predicted by YOLOv8 in OEP are fed to the ODP for accurate recognition. If Do={O1,O2,.,.,On} are the set of detected objects for an image; VGG16 is trained to recognize Ox∈Do where LSTM to extract the exact label lx∈Lo where Lo={l1,l2,.,.,ln}. In the decoding process, the annotation- A={Do,Lo} is used for predicting labels for the detected object and generating meaningful descriptive text. The working of the phase is presented in Fig. 7.

Figure 7 Schematic of the ODP from preprocessing to generation of semantic caption.

Encoding in ODP

For encoding the VGG16, introduced by Simonyan & Zisserman (2014) is exploited. The model comprises 13 convolutional layers and three fully connected layers. To these layers, a 3×3 filter is applied to capture details of an input image. Moreover, the feature map is reduced by using recurrent stacks of convolutional layers, which are then followed by a set of max-pooling layers. If m×m is the filter size where m=3, r and s the spatial position in an input image I with weight w and bias b, the convolution operation ( γ) of VGG16 is given as,

(23) γ(r,s)=∑i=1m⁡∑j=1m⁡I(r+i)(s+j).wi,j+b.

To take the maximum value from the pooling window, the max-pooling (MP) operation of VGG16, with i,j being the indices representing the offsets pooling window, is given in Eq. (24):

(24) MP(r,s)=maxi,j∈{0,1}⁡I(r+1,s+j).

The feature map Fm, where Fm=VGG16(I) is employed for onward processing. With a sliding window over Fm, region proposals Rp={r1,r2,.,.,rk} is generated, whereas the probability (P) that a region contains an object or part of an object is given as:

(25) P(ri|Fm)=σ(wt,Fm,br)

where σ is the sigmoid function, wt and br are the trainable weights. With the softmax function, a relevant class c is assigned to region- ri as:

(26) P(c|ri)=es∑i=1C⁡esi

where s is the score of class c and C is the number of classes.

An appropriate class ( Cp) for ri is predicated on having the maximum probability for Cp:

(27) Cp=argmaxcP(c|ri).

For nested objects, the attention weight ATij for region i, and j are computed from the expected feature maps ( Fmi, Fmj) as:

(28) ATij=(Fmi,Fmj)∑m=1P⁡exp(Fmi,Fmj).

In the proposed system, a resized image I of dimension 224×224×3 is utilized for VGG16. The image pixels are normalized in [0,1] before training, and excluding the fully connected layers, I is fed to the convolutional layer of VGG16 to extract Fm of dimension d; Fm∈Rd.

Decoding in ODP

As LSTM is an improved RNN with backpropagation (Hochreiter & Schmidhuber, 1997), the model is utilized for generating descriptive text for the detected objects. In the hidden layer, LSTM consists of interconnected memory cells. An input gate controls the input to a memory cell, while an output gate manages the output from a memory cell to the network. The model maps an input sequence X=(x1,x2,…,xn) to an output sequence Y=(y1,y2,…,yn) By iteratively applying unit activation using the following equations.

(29) It=χ(ωi[τt−1,xt]+βi)

(30) Ft=χ(ωf[τt−1,xt]+βf)

(31) Ot=χ(ωO[τt−1,xt]+βO)

where I, F and O are input, forget and output gates. ω is the weight matrices (e.g., ωi is the matrix of weights from the input gate and ωf, ωO are weight matrices for forgetting and output neurons, respectively). τt−1 represents the previous state of LSTM at time stamp t−1, whereas βx represents bias for the respective gates. The network output activation functions (Softmax) are represented by χ. Although LSMT has several versions, LSTM’s simple architecture consists of a memory cell and three gates (input, output, and forget) to regulate the flow of knowledge. The basic structure of a simple LSTM is shown in Fig. 8.

Figure 8 The standard structure of the LSTM model.

In the ODP, LSTM is trained by the caption of the image with the object class names. The model serves as a decoder to generate semantic text based on the features extracted by the encoder. At the text level, captions are tokenized into subwords. The vocabulary is transformed into sequences of integer indices. The label of objects- L={w1,w2,….,wn} is tokenized, followed by embedding for LSTM operation.

(32) LT={t1,t2,….tn}T∈Zn

The tokens are embedded into dense vector Dv of dimension k,

(33) Dv={e1,e2,…en}Dv∈Rk

Thus, the training data consists of image features, tokenized sequences, and embedded captions, as shown in Fig. 8.

The softmax activation function of LSTM computes probability distribution ‘ Pd’ for the prediction of a word- wxi to be in the un-normalized log probability U as:

(34) Pd(y=wxi|U)=eUi∑c⁡eUc

where U is the network’s output prior to applying softmax and c is the corresponding class. The temperature ( Tmp) of LSTM is kept moderate to avoid perspective twists in the production of text. With the inclusion of ‘ Tmp’, the function of softmax is altered as:

(35) Pd(y=wxi|U)=eUiTemp∑c⁡eUcTemp.

A 10% portion of the dataset was reserved for training and used as a validation split while the models were being trained. Such splitting means that the models learn from the training data without overlearning, so they can do well with unseen information. Every time a model is trained on the training data (80% of the set), it is tested on the validation data to check its accuracy, precision, recall and F1-score. We store the model that performed best on the validation set and test it further. Additionally, k-fold cross-validation ensures that our model is robust and the outcomes are not biased by a single data partition. After training, the models are tested to determine if any updates are needed, and this process is repeated 120 times in the example outlined. During text prediction from the feature map Fm and sequences of tokens- L, LSTM generates term wx at each hidden state ( hs). Each next term wx+1 is appended with the predicted text, where the process is repeated till the generation of a meaningful caption for the image (see Fig. 9). Equations representing the working of hs and next term are presented as;

(36) hs=LSTM(hs−1,[wx,Fm])

(37) wx+1=argmax(P(wx+1|,hs)).

Figure 9 Image features with tokenized and embedded captions.

Implementation and evaluation

The proposed framework is implemented in Python using the Ultralytics package in Colab. PyTorch and TensorFlow libraries are used for training, and NLTK is used for text tokenization and preprocessing. The Matplotlib and OpenCV are exploited for visualization and image manipulation.

Dataset

A custom dataset, PreEduDS, comprising 730 images, is created locally from online teaching resources and textbooks. Each of the images contains objects, letters, digits, and/or shapes normally taught at the preschool level. Besides the images folder, the captions are stored in a separate CSV file. The file is used for tokenization (using NLTK Punkt) and one-hot encoding. The W2V model is trained using the vocabulary dictionary with starting and trailing words as [‘<start>’, ‘Two’, ‘cats’, ‘with’, ‘digit’, ‘2’, ‘<end>’]. Besides the images and caption.csv, the labels and class names are also included in the dataset, as shown in Fig. 10.

Figure 10 Illustration of object recognition and textual description in ODP.

Training with validation

The system is trained using 80% of the images from the PreEduDS dataset, whereas it is validated and evaluated using 10% of each of the datasets. In the training of YOLOv8, images and label files are used, along with the classes. names file with the annotated images and captions for the training of VGG16 and LSTM. For effective training, validation is performed in parallel, with epochs set to 120 and a batch size of 8 in each. The history of the first 200 epochs is presented in Fig. 11.

Figure 11 Structure of the PreEduDS custom dataset.

Evaluation of the OEP

The first phase is evaluated by assessing how accurately YOLOv8 detects objects in both training images and unseen images from the test split. As in most preschool education software, teaching is based on pre-defined and built-in image datasets, which is why the train split is also included in the evaluation. For YOLOv8, a promising accuracy of 99.2% is achieved on the training set, whereas 93.7% is completed on the test split, resulting in an average accuracy of 96.4%. Precision, recall and F1-score obtained for the model are shown in Table 1, whereas the area under curve (AUC) is shown in Fig. 12. The formulas for precision (P), recall (R) and F1-score are given in the following equations.

(38) P=TPTP+FP

(39) R=TPTP+FN

(40) F1=2×P×RP+R.

Table 1 Outcomes of the evaluation of OEP.

Image source	AUC	Precision	Recall	Accuracy	F1-score	
Training	0.97	0.98	0.96	0.992	0.97	
Testing	0.92	0.93	0.91	0.937	0.94	

Figure 12 The cumulative accuracy-vs-epochs plot of the models.

If α represents the number of correct classifications and β the total number of predicted classes, accuracy A is given as:

(41) A=αβ.

Similarly, if p and q are the coordinates of the points on the curve, then AUC is represented as:

(42) AUC≈∑j=1j−1(pj+1−pj)2×(qj+qj+1).

As clear from Fig. 12, the quick raise at the top-left indicates the efficient performance of YOLOv8.

Evaluation of the ODP

The performance of the two models, VGG16 and LSTM, is assessed in terms of speed and accuracy using the same dataset splits. Although VGG16 is reported to have comparatively low accuracy for object recognition, LSTM generates captions with a promising accuracy of 96.5%. The AUCs of VGG16 and LSTM are shown in Fig. 13, while their precision, recall, and F1-scores are presented in Table 2.

Figure 13 The area under ROC curve about the performance of YOLOv8.

Table 2 Outcomes of the evaluation of ODP.

Model	Image source	AUC	Precision	Recall	Accuracy	F1-score	
VGG16	Training	0.94	0.9	0.91	0.96	0.96	
Testing	0.88	0.87	0.85	0.89	0.90	
LSTM	Training	0.98	0.93	0.97	0.98	0.94	
Testing	0.94	0.91	0.93	0.95	0.92	

The performance of the models is also cross-checked against other state-of-the-art models (Li et al., 2025; Kriouile et al., 2024; He et al., 2017; see Table 3). As shown in Fig. 14, a promising accuracy can be achieved when the number of epochs reaches 200. Therefore, assigning a value below 500 for epochs may further enhance the inference time, in addition to the other metrics. To check the accuracy of our results more thoroughly, we found the 95% confidence intervals for object recognition and text generation. 94.4% of the time, objects were correctly identified and the confidence interval suggests that number could be as high as 95.6%. With a 95% confidence interval between 95.2% and 97.8%, text generation was accurate to 96.5%. Paired t-test was applied to see if differences were statistically significant and results with p-values below 0.05 were considered so.

Table 3 Comparison of the three models with other standard models.

Model	Inference time (ms)	Params (M)	Memory (MB)	FLOPs	MACs	
Mask region-based CNN (Faster R-CNN) (Mask R-CNN) (Kriouile et al., 2024)	99	41	168	341	170	
Faster region-based CNN (Faster R-CNN) (He et al., 2017)	82	31	121	230	115	
Feature pyramid network (FPN) (Wang et al., 2025)	181	72	289	571	285	
YOLOv8 (Chen et al., 2025)	15	7.8	31	17.1	77	
VGG16 (Simonyan & Zisserman, 2014)	93	121	328	13.2	90	
LSTM (Hochreiter & Schmidhuber, 1997)	12	1.9	7	1.4	13	

Figure 14 The area under ROC curve of (A) VGG16 and (B) LSTM.

Conclusion and future work

Elementary and early childhood education has a significant role in the life-long cognitive and socio-emotional development of an individual. Visuals and images illustrating shapes and objects have a crucial role in childhood education, not only helping young learners to understand patterns with spatial relationships but also providing a strong foundation for math and problem-solving skills. The dedicated preschool illustrative software enables instructors to present content in a preferred manner and at a suitable pace. Moreover, young learners are engaged in an immersive way by fostering curiosity and creativity in learning. In all such visually engaging software, proper detection and recognition of objects are needed. Although several AI-based gamified software have been proposed, little attention has been paid to the precise detection and description of objects. As emerging deep learning technologies possess immense potential to recognize accurately and precisely present objects in input images, they are well-suited for this purpose. This research work presents a practical framework for accurately detecting objects and nested objects in an input image and precisely describes their contents. YOLOv8 and VGG16 are utilized for object detection and recognition, whereas the efficient LSTM model is used for predicting descriptive text. The framework is implemented in Colab using Ultralytics, PyTorch, OpenCV, and NLTK packages. A custom dataset, PreEduDS, containing 730 images, is used for training and testing. Accuracy scores of 94.4% and 96.5% are achieved for object detection and descriptive text generation, respectively. The comparative analysis, focusing on processing costs and resource utilization, demonstrates that the framework applies to emerging preschool tutoring software. The framework can be applied in interactive preschool apps, enhancing vocabulary by recognizing objects and generating descriptive text. It can also be integrated into AR tools, allowing children to interact with real-world objects. Additionally, it could assist educators by automatically labeling and describing objects in classroom materials. As part of our future work, the framework will be enhanced to pronounce the detected objects and present descriptive text in various natural languages.

Supplemental Information

Supplemental Information 1 Code.

Additional Information and Declarations

Competing Interests

The authors declare that they have no competing interests.

Author Contributions

Lan Lv conceived and designed the experiments, performed the experiments, analyzed the data, performed the computation work, prepared figures and/or tables, authored or reviewed drafts of the article, and approved the final draft.

Suhui Yao conceived and designed the experiments, performed the experiments, analyzed the data, performed the computation work, prepared figures and/or tables, authored or reviewed drafts of the article, and approved the final draft.

Data Availability

The following information was supplied regarding data availability:

The code is available in the Supplemental File.

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
