# Peer review of "A robust detect and describe framework for object recognition in early childhood education"

_PeerJ Computer Science, doi:10.7717/peerj-cs.3080_

## Round 0.1 · original submission · Major Revisions

Dear authors thank you for your submission. Based on the input from experts I would like to inform you that your manuscript is not acceptable in its current form and needs couple of improvements as mentioned in their comments. Please carefully incorporate those also mine below and resubmit.

AE comments

1. The dataset description could be improved by offering more details on the image types used, their source, and any potential biases in the dataset.
2. Introduction section needs to be expanded
3. The literature review is concise but could benefit from further expansion. It would be helpful to include more studies on the application of emerging technologies like AR, VR, and deep learning in early education.
4. The formatting of equations, particularly in Equation (1), could be improved for clarity. Ensure that all variables are defined and the equations are formatted correctly.
5. Figure 1 is referenced as the schematic for the framework but is not sufficiently explained in the text. It would be helpful to include a brief explanation of the components shown in the figure.
6. The framework's potential applications in preschool education are briefly mentioned in the conclusion, but specific examples of how it could be implemented in real-world settings would make the conclusion more impactful.
7. The paper discusses the use of Yolov8, VGG16, and LSTM for detection and description tasks. A direct comparison of these models' strengths and weaknesses could be valuable for readers to understand the model selection process.

**Language Note:** PeerJ staff have identified that the English language needs to be improved. When you prepare your next revision, please either (i) have a colleague who is proficient in English and familiar with the subject matter review your manuscript, or (ii) contact a professional editing service to review your manuscript. PeerJ can provide language editing services - you can contact us at [email protected] for pricing (be sure to provide your manuscript number and title). – PeerJ Staff

Reviewer 1 ·

Basic reporting

.

Experimental design

.

Validity of the findings

.

Additional comments

The paper titled "A Robust Detect and Describe Framework for Object Recognition in Early Childhood Education" presents an interesting framework for object recognition and description in preschool education. After reviewing the paper, I have some concerns which must be addressed to make it better fit for publication.
1. In the abstract, the sentence "Understanding the basic shape, colors, and letters at the early age laid a strong foundation for academic excellence and emotional growth" should be rephrased as: "Understanding basic shapes, colors, and letters at an early age lays a strong foundation for academic excellence and emotional growth."
2. In Section 3 (Methodology), the dataset PreEduDS is briefly mentioned but lacks sufficient detail on how it was curated. Additional explanation about its sources would help clarify the dataset's relevance and quality.
3. In sub section 3.2, the paper uses the term "feature pyramid network" in relation to Yolov8, but this term is not explained. It would be helpful to define it or provide a reference for further reading.
4. While Non-Maximum Suppression (NMS) is introduced, its explanation could be expanded to clarify how it improves object detection accuracy by removing redundant detections.
5. The interaction between the Yolov8, VGG16, and LSTM models is not clearly explained. A more detailed description of how these models work together within the framework would provide clarity.
6. The paper mentions accuracy scores (94.4% for object recognition, 96.5% for text generation) but does not include statistical significance or confidence intervals. Including this data would strengthen the validity of the results.
7. In the methodology section, the paper could benefit from a more detailed explanation of how the training and evaluation processes for the models are structured. For example, how are the models validated during training?
8. The section discussing the LSTM model could be expanded to provide more information on how it generates descriptive text and integrates with the object recognition phase.

·

Basic reporting

The manuscript is written in a structured manner, with a logical flow across sections. The figures and tables are relevant and well-annotated. However, the manuscript contains several grammatical errors and awkward sentence constructions that make some parts difficult to read. For example, phrases like "laid a strong foundation" (should be "lays"), or "perfectly detected" (better as "detected with high accuracy") can be improved. It is strongly recommended that the manuscript undergo a comprehensive language edit to enhance clarity and professional tone. Literature review is adequately referenced and gives good context, though recent work using transformer-based image captioning models could be briefly acknowledged.

Experimental design

The research objective is clearly stated, focusing on enhancing preschool learning through an object detection and captioning system. The experimental design—comprising object detection (YOLOv8) and object description (VGG16 + LSTM)—is well structured and described in sufficient detail to support reproducibility. The use of a custom dataset (PreEduDS) is appropriate for the domain. However, details about the dataset split (e.g., randomization, stratification) and rationale for choosing specific models (e.g., VGG16 over more modern CNNs or transformers) should be elaborated. An ablation study to understand the contribution of each component would further strengthen the methodology.

Validity of the findings

The results are promising, with reported accuracies of 94.4% (YOLOv8) and 96.5% (LSTM), and are supported by evaluation metrics such as AUC and F1-score. The manuscript includes a useful performance comparison against other models. However, the dataset size (730 images) is relatively small, which may limit generalizability. The authors should mention this limitation and discuss future plans to expand or validate their model on public datasets. Including qualitative examples (e.g., detected objects with predicted captions) would improve the impact and help readers assess practical applicability.

Additional comments

The authors have presented a strong technical contribution to the intersection of deep learning and early education. The division into OEP and ODP phases is intuitive and well justified. Future directions, such as integrating multilingual speech outputs, are promising. The paper would benefit from minor revisions addressing language, clearer justification of model choices, and some deeper methodological clarifications. With these addressed, the work is suitable for publication.

---

## Round 0.2 · accepted · Accept

Dear authors,
Based on the input from experts on the revised version and my assessment, I am pleased to inform you that your manuscript is being considered scientifically sound for recommendation towards acceptance.
Thank you for your valuable contribution, and I wish you good luck with your future research.

Reviewer 1 ·

Basic reporting

All the comments have been addressed by the authors.

Experimental design

All the comments have been addressed by the authors.

Validity of the findings

All the comments have been addressed by the authors.

Additional comments

All the comments have been addressed by the authors.